# Identifying Rural Landscape Heritage Character Types and Areas: A Case Study of the Li River Basin in Guilin, China

Zizhen Hong [1,2], Wentao Cao [2], Ying Chen [2], Sijia Zhu [2] and Wenjun Zheng [2,*]

1    College of Landscape Architecture and Horticulture Science, Southwest Forestry University, Kunming 650224, China; hongzizhen@hotmail.com

2    College of Tourism and Landscape Architecture, Guilin University of Technology, Guilin 541006, China; cwt_1999@163.com (W.C.); chenying6990@163.com (Y.C.); zsjguilin@163.com (S.Z.)

*    Correspondence: 2004023@glut.edu.cn

**Abstract:** Rural landscape heritage faces issues of landscape character homogenization and unclear protection boundaries. We propose combining landscape character assessment (LCA) methods to identify the characteristics and areas of heritage, aiming to preserve the diversity and integrity of the landscape. This paper focuses on the Li River Basin as the study area, presenting a method for identifying characteristics and areas of rural landscape heritage. It is divided into four steps: selection and spatial scope identification of rural landscape heritage, identification of natural character areas, identification of cultural character areas, and identification and analysis of character areas of rural landscape heritage. Firstly, cultural relic units, traditional villages, and intangible cultural heritage as sources of rural landscape heritage were selected by utilizing the Minimum Cumulative Resistance model (MCR) to calculate the spatial scope of rural landscape heritage. Secondly, clustering and automatic partition methods were employed to classify the Li River Basin into four types of natural character areas. Thirdly, cultural core areas and buffer areas were determined based on the heritage source hierarchy and cultural features. Fourthly, by overlaying heritage spatial ranges, natural character areas, and cultural character areas, 2 levels of heritage areas, 7 types of heritage cultural areas, and 43 heritage character units were obtained. This method not only provides a comprehensive framework for the identification of characteristics and areas for rural landscape heritage but also enhances the integrity of data selection in landscape character assessment methods at the cultural level.

**Keywords:** rural landscape heritage; landscape character assessment; Li River Basin

## 1. Introduction

Rural areas represent one of the most harmonious human habitats and spaces formed during the co-evolution of humans and nature. They constitute a complex organic system that balances natural conservation, human survival, and social development [1]. Rural landscapes are a crucial component of human heritage, providing social and economic benefits, ecosystem services, and diverse cultural support [2]. Since the formulation of the Venice Charter on the Conservation and Restoration of Monuments and Sites (1964) and the release of the Guidelines on Rural Landscape as Heritage by the International Council on Monuments and Sites (ICOMOS) and the International Federation of Landscape Architects (IFLA) in 2017 [3,4], rural landscapes and their heritage have become significant focuses of international heritage conservation [5,6]. Rural landscape heritage faces challenges of homogeneous landscape characteristics and unclear protection boundaries. On the one hand, the pressures and threats on rural areas intensify due to urbanization and industrialization, leading to the gradual loss of local characteristics and diverse cultures within rural landscapes [7–10]. On the other hand, rural landscapes constitute complex landscape systems [11], requiring comprehensive planning and zonal controls due to the diverse

element types and intricate spatial distribution characteristics they possess. Currently, many scholars focus on macro-corridor planning and micro-entity conservation centered around heritage sources, yet a systematic zoning protection method for rural landscape heritage has not been established [12–16]. To preserve the uniqueness and diversity of rural areas, maintain the spatial patterns and local culture, and support sustainable rural development, it is crucial to identify and zone the characteristics of rural landscape heritage systematically. Therefore, a systematic approach to identifying heritage characteristics and delineating conservation areas is needed.

Landscape Character Assessment (LCA) has the ability to identify and assess factors within a specific area that make the landscape distinctive, creating a unique sense of place [17]. Landscapes are the result of the interaction between nature and culture [18]; therefore, integrating natural and cultural elements is indispensable in landscape character recognition. LCA guidelines offer a systematic inventory, primarily categorized into natural and cultural elements [17]. With the open-source availability of geographic information data, obtaining natural data is relatively easy and comprehensive, supporting a systematic data structure [19]. However, cultural element types and content complexity have not yet formed a systematic element system incorporated into the overlay analysis of character recognition. Certainly, some studies incorporate cultural characteristics such as ethnic populations, settlement culture, and architectural form into the recognition results at the cultural level [20–22]. However, overall, the selection of cultural data types tends to be relatively narrow, limiting the systematicity and reliability of LCA in the identification process. Strengthening the systematic selection of cultural data during the identification process enhances the scientific validity of the methods and results of landscape character recognition. Rural landscapes, as cultural heritage, represent the most widespread and representative cultural landscapes. They are spaces where various landscape types interweave and coexist, encompassing not only traditional villages and historical relics but also involving traditional knowledge and techniques related to the relationship between humans and nature. The collaborative interaction of these elements forms a crucial foundation for the harmonious coexistence of humans and nature [4]. Therefore, rural landscape heritage serves as a valuable supplement to cultural data.

Utilizing landscape character recognition as a method for zoning and protective planning of rural landscape heritage is a reasonable choice. On one hand, from a spatial management perspective, although the establishment and improvement of China's natural conservation areas and heritage protection system can effectively safeguard significant natural and cultural regions [23], there are still many heritage sources not within the management scope. Consequently, this leads to an overemphasis on the core areas within the protected zones, overlooking high-value heritage sources outside these zones, and undermining the connections between protected zones and their surroundings. Viewing the landscape characteristics within the region from a macro perspective is beneficial for integrating the complex and fragmented distribution of heritage within the space, thereby strengthening the overall management of regional protection. On the other hand, from a spatial relationship perspective, there are numerous high-quality cultural landscapes both within and outside the protected zones. Identifying and zoning resources with similar characteristics, and analyzing the distribution characteristics and cultural attributes of heritage in different characteristic zones [24] can assist decision-making bodies in establishing a systematic understanding of landscape heritage characteristics and developing comprehensive protection plans.

The Li River Basin is located within the jurisdiction of Guilin City, Guangxi, China, with two important protected areas and 221 major material heritage sources. The two important protected areas include the "Southern China Karst" World Natural Heritage Site and a national scenic area in China. The 221 material heritage sources comprise 135 cultural relic units and 86 traditional villages at various administrative levels. From the distribution of protected areas and heritage sources, it is evident that 160 material heritage sources in rural areas are not within the protected zones, lacking attention and protection (Figure 1).

With the rapid advancement of urbanization and the thriving development of tourism, the basin faces issues such as environmental degradation and the weakening of landscape characters. To monitor and protect the ecological environment of the basin, researchers and managers maintain a high level of attention. On the one hand, researchers focused on land use and landscape pattern change [25–28], ecosystem services and value estimation [29,30], and found that the area of cultivated land, water body and grassland within the basin decreased, the area of construction land and bare land increased, and the value of ecosystem services gradually decreased. The sustainable development of the ecological environment of the river basin faces challenges. On the other hand, government departments have formulated corresponding protective regulations and planning projects, including the "Guangxi Zhuang Autonomous Region Li River Basin Ecological Environment Protection Regulations", "Guilin City Li River Scenic Area Management Regulations", and "Guangxi Li River Ecological Comprehensive Governance Demonstration Project" [31–33]. These initiatives provide sustainable protection and management for the Li River Basin. In summary, based on the significant value and environmental challenges faced by the Li River Basin, researchers and managers have maintained continuous attention and scientific support for the natural ecological resources of the basin. However, there is a lack of exploration and protection of humanistic resources. Therefore, using the Li River Basin as a case study, this paper applies the Minimum Cumulative Resistance model to calculate the spatial extent of rural landscape heritage. It utilizes a landscape character identification method to recognize the natural and cultural character zones of heritage spatial extent, followed by overlay and analysis to obtain heritage character zones. It is hoped that this method can promote comprehensive protection of rural landscape heritage, contributing to the sustainable preservation and utilization of rural landscapes in the Li River Basin.

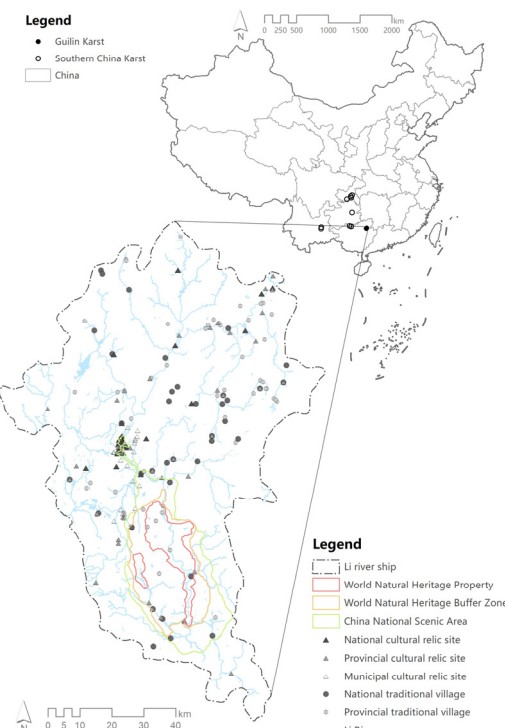

**Figure 1.** Distribution map of important protected areas and major material heritages in the Li River Basin.

## 2. Materials and Methods

### 2.1. Study Area

The Li River basin is located within the northeastern part of Guilin City in the Guangxi Zhuang Autonomous Region, China (Figure 2). It belongs to the Xi River system, a major tributary of the Pearl River basin. Originating from the wetlands of Maor Shan, the main

peak of Yuecheng Ridge in the northwest of the Nanling Mountains, the Li River flows through Pingxiang Town in Pingle County, where it converges with the Lipu River and the Gongcheng River to form the Gui River. The basin includes 19 tributaries, such as Taohua River, Xiaodong River, Nanxi River, and Xiangsi River, with a total length of approximately 300 km and an area of about 8100 km². The Li River basin boasts rich landscape resources, encompassing natural heritage resources with "green mountains, clear waters, unique caves, and beautiful rocks", as well as vibrant and diverse cultural heritage resources. These elements are interdependent, influencing each other and merging to nurture unique, extraordinary, and picturesque landscapes, embodying multiple values in terms of nature, culture, society, research, and history.

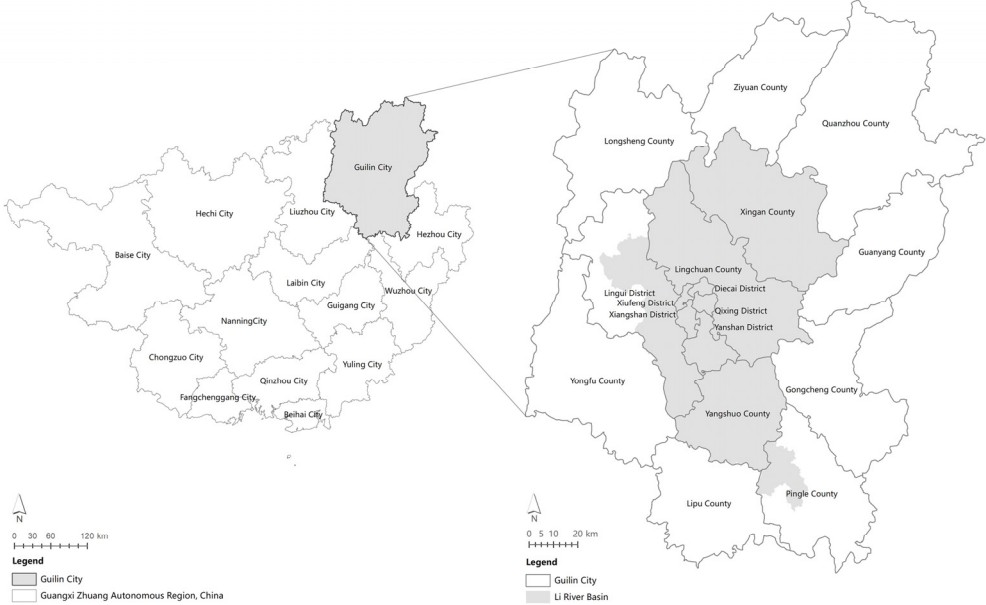

**Figure 2.** Study area.

### 2.2. Data Sources

This study involves two categories of data: natural resource data and rural landscape heritage data. Firstly, for natural landscape data, elevation, terrain undulation, and land cover are selected as landscape character factors. Utilizing Cloud DEM data in ArcGIS, the elevation and terrain undulation within the basin are extracted and calculated. The land cover data are sourced from the fine classification product of the global 30 m land surface cover released by the Aerospace Information Research Institute of the Chinese Academy of Sciences in 2020 [34]. Secondly, rural landscape heritage data comprise cultural relics protection units, traditional villages, and intangible cultural heritage. For cultural relics' protection units, a total of 138 heritage points are obtained by overlaying national-, provincial-, and municipal-level cultural relics' protection units with the basin's scope. Traditional villages are derived from the list of traditional villages in China and Guangxi, resulting in 87 points after intersection with the Li River basin in ArcGIS. Intangible heritage includes 88 data points based on national-, provincial-, and municipal-level intangible cultural heritage lists.

### 2.3. Research Methods

Landscape Character Assessment (LCA), as a perceptual tool integrating natural and cultural landscape characteristics [35], stands as a crucial method in planning and managing landscape areas [36–38]. The identification process of LCA can be categorized into holistic and parameter methods [39]. The holistic approach relies on comprehensive human cognition, incorporating experiential and subjective judgments. In contrast, the parameter method involves analyzing various types of geographic data to form a map of landscape

characteristics [39–41], employing techniques such as clustering, overlay, and automatic partitioning [19,40,42]. While the parameter method offers greater objectivity, the quality of identification results heavily depends on the systematic and comprehensive selection of data types [9]. Despite the greater objectivity of the parameter method compared to the holistic approach, the quality of identification results significantly relies on the systematic and comprehensive selection of data types.

In response to the challenges of landscape character homogeneity and unclear protection areas faced by rural landscape heritage, this paper introduces a methodology that combines subjective overall assessment and objective parameters based on LCA to establish a recognition process for the characters and regions of rural landscape heritage. LCA emphasizes the holistic perception of natural and cultural elements, and rural landscape heritage represents an integration of both natural and cultural aspects. Therefore, building upon a clear delineation of heritage spatial extent, this study identifies the natural and cultural character areas of the heritage. These are integrated to form the character zones of rural landscape heritage, providing insights for regional planning and management and assisting in the sustainable preservation and utilization of rural landscapes.

The research is divided into 4 steps, encompassing the selection and spatial scope of rural landscape heritage, identification of natural character areas, identification of cultural character areas, and identification and analysis of rural landscape heritage character areas (Figure 3). Initially, cultural relic units, traditional villages, and intangible cultural heritage are chosen as sources of rural landscape heritage. The Minimum Cumulative Resistance model (MCR) is then employed to calculate the spatial extent of rural landscape heritage. Secondly, clustering and automatic partitioning methods are applied to delineate natural character zones within the Li River Basin. Thirdly, the area is further divided into cultural zones based on heritage source levels and cultural characters. Lastly, through the overlay of heritage spatial scope, natural character areas, and cultural character areas, the character areas of rural landscape heritage are formed.

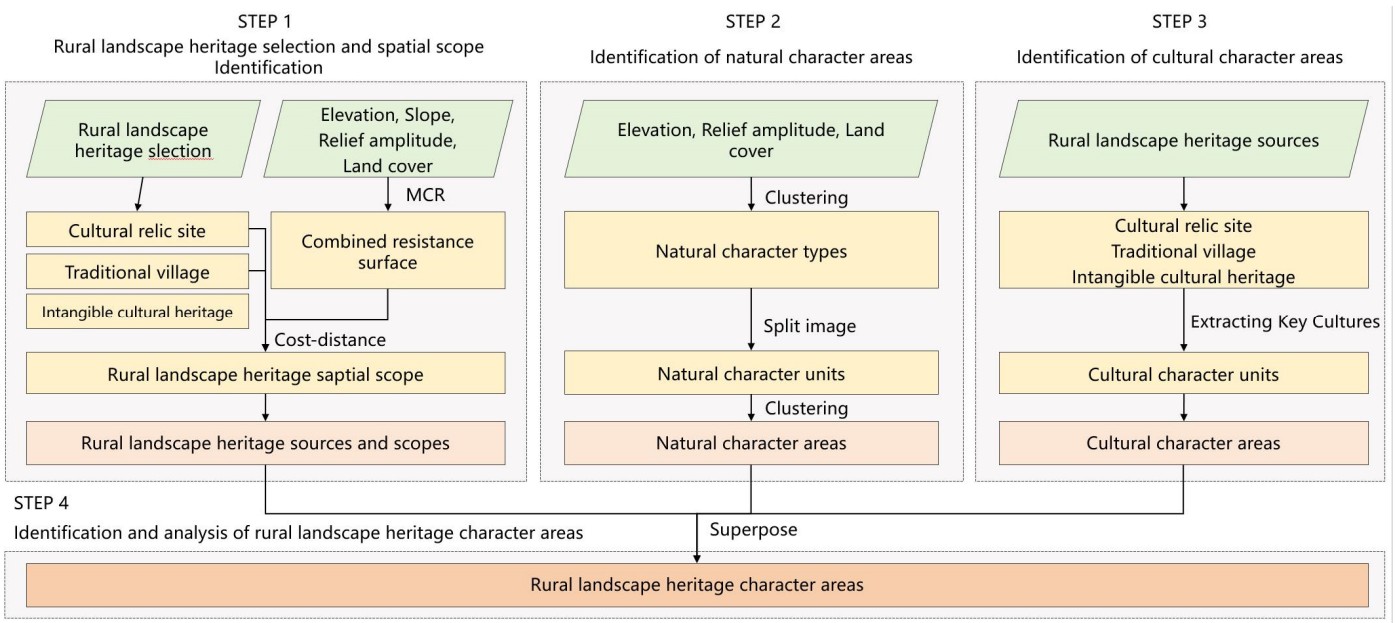

**Figure 3.** Rural landscape heritage characterization and area identification process.

2.3.1. Selection and Spatial Scope Identification of Rural Landscape Heritage

This research is based on the National Heritage Protection List and field investigations. Cultural relic sites, traditional villages, and intangible cultural heritage are selected as sources of rural landscape heritage. The Minimum Cumulative Resistance (MCR) model is utilized to calculate the spatial extent of heritage. Currently, the MCR model, originally

widely applied in habitat isolation assessments, has found extensive use in ecological security pattern evaluations and landscape heritage corridor planning [43–47].

Initially, elevation, slope, relief amplitude, and land cover are chosen as resistance factors. The resistance values for these factors are determined by referencing existing relevant studies, professional advice, and the distribution of heritage within the study area. Subsequently, resistance surfaces for each factor are computed, and a comprehensive resistance surface is established by combining the weighted resistance surfaces using the Analytic Hierarchy Process (AHP) [48]. Finally, the spatial extent of rural landscape heritage is calculated based on the MCR model and cost–distance relationship. Although intangible cultural heritage is closely connected to cultural relic sites and traditional villages, its spatial location is challenging to determine, and therefore, it is not considered in the calculation of spatial extent recognition.

### 2.3.2. Identification of Natural Character Areas

Natural characteristics include elevation, relief amplitude, and land cover, three variables with the greatest impact on the surface landscape. These variables are imported into ArcGIS, with a unified coordinate system, spatial resolution, and grid unit for each scale. A spatial resolution of 30 m and a grid unit of 0.2 km × 0.2 km are chosen. The matrices' variables and grid units are extracted and imported into SPSS 25. Two-step clustering is employed to obtain landscape character types. Subsequently, eCognition 9.0 image software is used to spatially segment the landscape character type map and obtain landscape character units. To reduce the fragmentation of landscape character types and enhance the recognition of regional characteristics, clustering algorithms are further applied to landscape units. The number of clusters is determined using the elbow method and silhouette coefficient. The clustering results are then mapped into ArcGIS 10.7 to obtain natural landscape character zones.

### 2.3.3. Identification of Cultural Character Areas

Cultural relic sites, traditional villages, and intangible cultural heritage are important categories listed in China's Heritage Protection Catalog, featuring a comprehensive evaluation system. For instance, traditional Chinese villages are assessed based on three aspects: architecture (including antiquity, rarity, scale, proportion, richness, integrity, and aesthetic value), layout (comprising antiquity, richness, layout integrity, scientific and cultural value, and coordination), and intangible cultural elements (involving rarity, richness, continuity, scale, vitality of inheritors, and dependence). Thus, rural landscape heritage can significantly reflect regional cultural characteristics.

Within China's heritage protection system, heritage is reported through various administrative levels, creating national-, provincial-, and municipal/county-level protection lists. National-level heritage generally holds higher comprehensive value, followed by provincial-level, with municipal/county-level heritage having the least impact. In this study, the spatial extent of national-level heritage sources is set as the cultural core area, while the remaining areas are designated as the cultural buffer zone. By integrating the cultural characters and content of cultural relic sites, traditional villages, and intangible cultural heritage, and based on the cultural core area, this study extracts corresponding key cultural elements, forming cultural character units and areas.

### 2.3.4. Identification of Heritage Character Areas

The heritage spatial scope, natural character areas, and cultural character areas are overlaid and adjusted to derive the character areas of rural landscape heritage. Using the naming convention "Natural Character Type Number—Cultural Character Unit Number", a heritage character zone map is created. By analyzing the spatial distribution and structure of heritage types within the regions, an understanding of the characters of rural landscape heritage is obtained.

2.3.5. Limitation of Research Method

The method for identifying rural landscape heritage characteristics and regions constructed in this study is generally applicable to macro-scale spatial regions and can be extended to research on heritage character identification and regional protection from other macro perspectives. However, it still has the following limitations: Although the study incorporates tangible and intangible heritage into the cultural elements of landscape character identification, due to the large scale of the research area, the manual field correction process is conducted by selecting representative points and areas. Furthermore, this paper establishes a data system at the cultural level focusing on rural landscapes, which may not be suitable for recognizing landscape characteristics in urban areas.

## 3. Results

### 3.1. Selection and Spatial Scope Identification of Rural Landscape Heritage

#### 3.1.1. Selection of Rural Landscape Heritage

The heritage of rural landscapes is intertwined in the same space, and it is numerous, complex and relational. In order to facilitate the interpretation and analysis of the scope and connotation of heritage, and to sort out the categorization composition and elemental relationships, it is necessary to establish a regional and holistic concept. The conservation of rural landscape heritage should be based on its constituent elements. Therefore, in order to completely reflect the types, relationships and synergistic mechanisms of each element of rural landscape heritage, based on the definition of rural landscape heritage in the document "Guidelines on Rural Landscape as Heritage", tangible heritage and intangible heritage are selected as the two major components of rural landscape heritage, including cultural protection units, traditional villages and intangible cultural heritage. A total of 309 heritage sources (135 cultural heritage units, 86 settlement heritage and 88 intangible heritage) were screened in the Li River Basin.

#### 3.1.2. Identification of Rural Landscape Heritage Spatial Scope

First, the resistance classification and resistance values for elevation, slope, terrain relief, and land cover were determined based on the trends in AP/NRLH values (Table 1). Regarding elevation, the decrease in AP/NRLH values with increasing elevation indicates that cultural dissemination and settlement construction gradually become more challenging with rising elevation. Similarly, in terms of slope and terrain undulation, the decrease in AP/NRLH values with increasing numerical values indicates a gradual increase in resistance to cultural influence. In terms of land cover, farmland and water bodies are crucial manifestations of human–land relationships, serving as essential links between forests and towns and being the most prevalent spatial types for rural heritage distribution. Hence, their resistance values are relatively low. Impervious surfaces represent spaces with concentrated human activities, enhancing heritage accessibility; hence, their resistance values are moderate. Spaces such as forests, shrubs, and wetlands sustain ecological habitats, resulting in a lower distribution of heritage, and consequently, higher resistance values.

**Table 1.** Grades and resistance values of the resistance factors.

| Resistance Factor | Weight | Grade | Area Percentage (AP,%) | Number of Rural Landscape Heritage (NRLH) | NRLH/AP | Resistance Value |
|---|---|---|---|---|---|---|
| Elevation (m) | 0.2966 | <200 | 29.502 | 129 | 4.373 | 50 |
| | | 200–500 | 44.132 | 86 | 1.949 | 100 |
| | | >500 | 26.366 | 6 | 0.228 | 500 |

**Table 1.** *Cont.*

| Resistance Factor | Weight | Grade | Area Percentage (AP,%) | Number of Rural Landscape Heritage (NRLH) | NRLH/AP | Resistance Value |
|---|---|---|---|---|---|---|
| Slope (°) | 0.0532 | <5 | 25.974 | 110 | 4.235 | 10 |
| | | 5–15 | 28.921 | 78 | 2.697 | 100 |
| | | 15–25 | 24.661 | 29 | 1.176 | 300 |
| | | >25 | 20.443 | 4 | 0.196 | 500 |
| Relief amplitude (°) | 0.1018 | <30 | 72.439 | 210 | 2.899 | 10 |
| | | 30–70 | 27.073 | 11 | 0.406 | 100 |
| | | 70–200 | 0.975 | 0 | 0.000 | 300 |
| | | >300 | 0.100 | 0 | 0.000 | 500 |
| Land cover | 0.5485 | Rainfed cropland; irrigated cropland; water body | - | - | - | 10 |
| | | Impervious surfaces; grassland; herbaceous cover | - | - | - | 50 |
| | | Open evergreen broadleaved forest; Closed evergreen broadleaved forest; Open deciduous broadleaved forest (0.15 < fc < 0.4); closed deciduous broadleaved forest (fc > 0.4); open evergreen needle-leaved forest (0.15 < fc < 0.4); closed evergreen needle-leaved forest (fc > 0.4) | - | - | - | 100 |
| | | Shrubland; evergreen shrubland; sparse vegetation (fc < 0.15) | - | - | - | 300 |
| | | Wetlands | - | - | - | 500 |

Second, the relative importance of the resistance factors for constructing the judgment matrix was determined based on the existing literature and field surveys, the importance of each resistance factor was ranked, and the scope of the heritage's cultural influence was calculated based on heritage sources. Importance ranking of each resistance factor was also conducted. The Analytic Hierarchy Process (AHP) was employed to calculate the weights using the judgment matrix, resulting in elevation, slope, terrain undulation, and land cover weights of 0.2966, 0.0532, 0.1018, and 0.5485, respectively. The comprehensive resistance surface was computed by combining the resistance surfaces of each factor with their respective weight coefficients (Figure 4). Subsequently, the MCR model was utilized to calculate the spatial scope for rural landscape heritage sources (Figure 5).

*3.2. Identification of Natural Character Types and Areas*

3.2.1. Natural Character Types

This study selected three character factors, elevation, relief amplitude, and land cover, to establish a grid within the Li River basin. Variables of these factors were extracted for second-order clustering. Using SPSS 25 software, the clustering quality value was calculated as 0.4, indicating good cohesion and separation with significant inter-group differences. Based on the clustering results and multiple attempts with different numbers of clusters, the best clustering number was determined to be four (Figure 6). The distribution characteristics were described and summarized by statistically analyzing the basic and relative distribution of each heritage type (Table 2). (1) In terms of basic characteristics, considering the mean, grid number, and area proportion of natural character types, type 3 has the largest area proportion, accounting for 42.6%. Types 1, 2, and 4 follow, with proportions of 25.4%, 16.3%, and 15.7%, respectively. (2) In terms of relative distribution, type 1's elevation is mainly concentrated below 500 m, relief amplitude shows a polarized distribution, and land type

consists of single rain-fed farmland. Type 2's elevation is also concentrated in the 0~500m range, with relief amplitude concentrated below 30°, and diverse land types including irrigated farmland, shrubs, water bodies, and impermeable surfaces. Type 3's elevation is concentrated in the 500~1000 m range, with relief amplitude greater than 30°, evenly distributed, and land type as closed evergreen broadleaf forest. Type 4 has relatively higher elevation and relief amplitude values, concentrated in the ranges above 1000 m and 30~90°, with evergreen and deciduous mixed forest land types. Based on the statistics and analysis, types 1 to 4 can be described as follows (Table 3): plateau and hilly rain-fed farmland area, urban farmland area with gentle terrain and mixed hilly topography, hilly terrain with evergreen dense forest, and mountainous terrain with mixed dense forest.

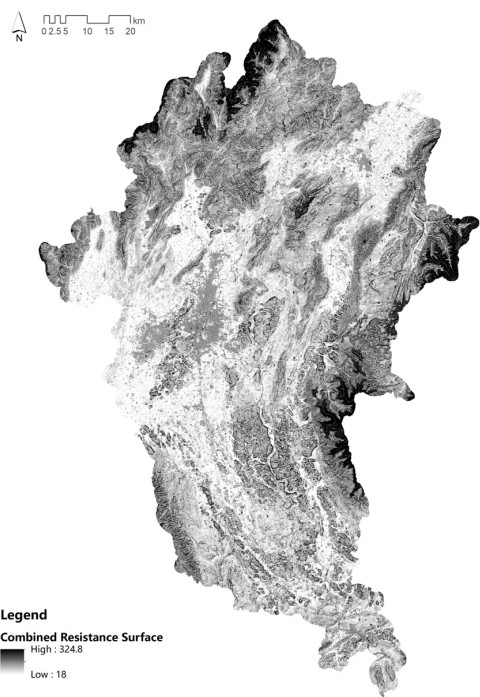

**Figure 4.** Combined resistance surface in the Li River Basin.

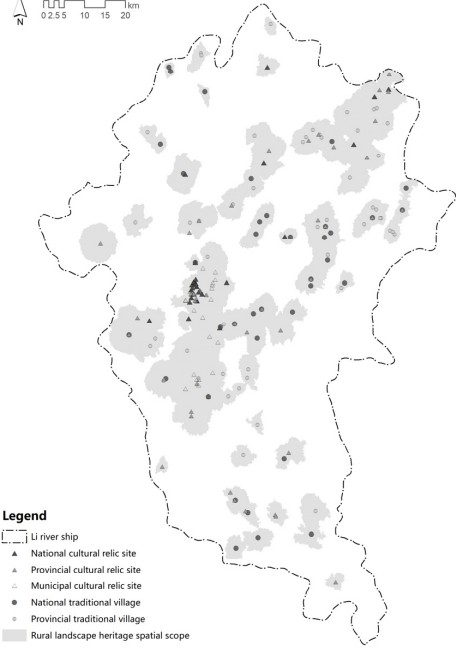

**Figure 5.** The spatial scope of rural landscape heritage in the Li River Basin.

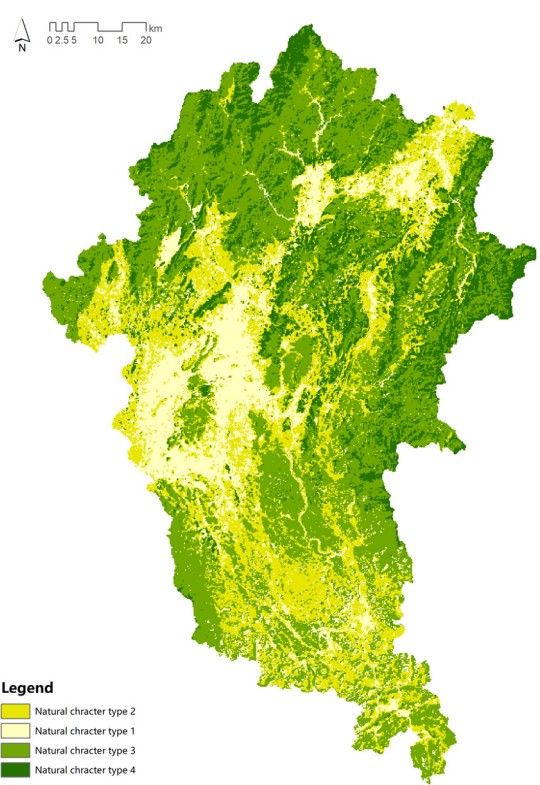

**Figure 6.** Distribution of natural character types in the Li River Basin.

**Table 2.** Statistics of natural character types in the Li River Basin.

| Cluster Variables | | Type 1 | Type 2 | Type 3 | Type 4 |
|---|---|---|---|---|---|
| Elevation | Average (m) | 220.19 | 186.25 | 491.68 | 639.94 |
| Relief amplitude | Average (°) | 10.83 | 8.77 | 28.92 | 30.13 |
| Land Cover | Rainfed farmland | 52,004 | 0 | 0 | 0 |
| | Irrigated farmland | 0 | 18,563 | 0 | 0 |
| | Open evergreen broadleaf forest | 0 | 0 | 0 | 1 |
| | Closed evergreen broadleaf forest | 0 | 0 | 87,234 | 0 |
| | Open deciduous broadleaf forest | 0 | 5 | 0 | 0 |
| | Closed deciduous broadleaf forest | 0 | 0 | 0 | 7156 |
| | Open evergreen mixed forest | 0 | 5 | 0 | 0 |
| | Closed evergreen coniferous forest | 0 | 0 | 0 | 24,921 |
| | Thicket | 0 | 1 | 0 | 0 |
| | Evergreen thicket | 0 | 4301 | 0 | 0 |
| | Impervious surface | 0 | 8391 | 0 | 0 |
| | Water body | 0 | 1860 | 0 | 0 |
| | Null | 0 | 281 | 0 | 11 |
| Total | Number of grids (count) | 52,004 | 33,407 | 87,234 | 32,089 |
| | Area (km$^2$) | 2080.02 | 1337.03 | 3489.34 | 1283.12 |
| | Percentage (%) | 25.40 | 16.30 | 42.60 | 15.70 |

**Table 3.** Description of natural character types in the Li River Basin.

| Natural Character Type | Type 1 | Type 2 | Type 3 | Type 4 |
|---|---|---|---|---|
| Elevation Range | 0~500 m | 0~500 m | 500~1000 m | >1000 m |
| Terrain Undulation Range | <30°, >90° | <30°, >90° | >30° | 30°~90° |

**Table 3.** *Cont.*

| Natural Character Type | Type 1 | Type 2 | Type 3 | Type 4 |
|---|---|---|---|---|
| Land Cover | Rainfed farmland | Irrigated farmland, thicket, water body, impervious surface | Closed evergreen broadleaf forest | Closed deciduous broadleaf forest, open evergreen mixed forest |
| Type Description | An area characterized by a combination of plateaus and hilly terrain, featuring rainfed farmland | An area with gentle terrain and mixed hilly topography, including urban and farmland | A region dominated by hilly terrain covered with evergreen dense forests | A region dominated by mountainous terrain covered with mixed dense forests |

### 3.2.2. Natural Character Zoning

Due to the fragmented nature of the identified natural character types, image segmentation and clustering methods were introduced for further zoning. Firstly, on the basis of landscape character types, the eCognition 25 software was used to segment regions. Ideal segmentation results were obtained with scale and color index values of 200 and 0.2, respectively, resulting in 547 units (Figure 7a). Secondly, the areas of segmented regions were statistically recorded to form a data matrix. The Elbow Method was used to determine the optimal clustering interval, which was found to be from 2 to 10 classes. Silhouette coefficients were then calculated for each clustering number (K) within the interval, revealing that the three, four, five, and seven classes had good clustering quality (Figure 8, Table 4). Finally, the results were mapped into ArcGIS to compare the clustering effects of each class, and four classes were determined to be the optimal regional clustering number (Figure 7b).

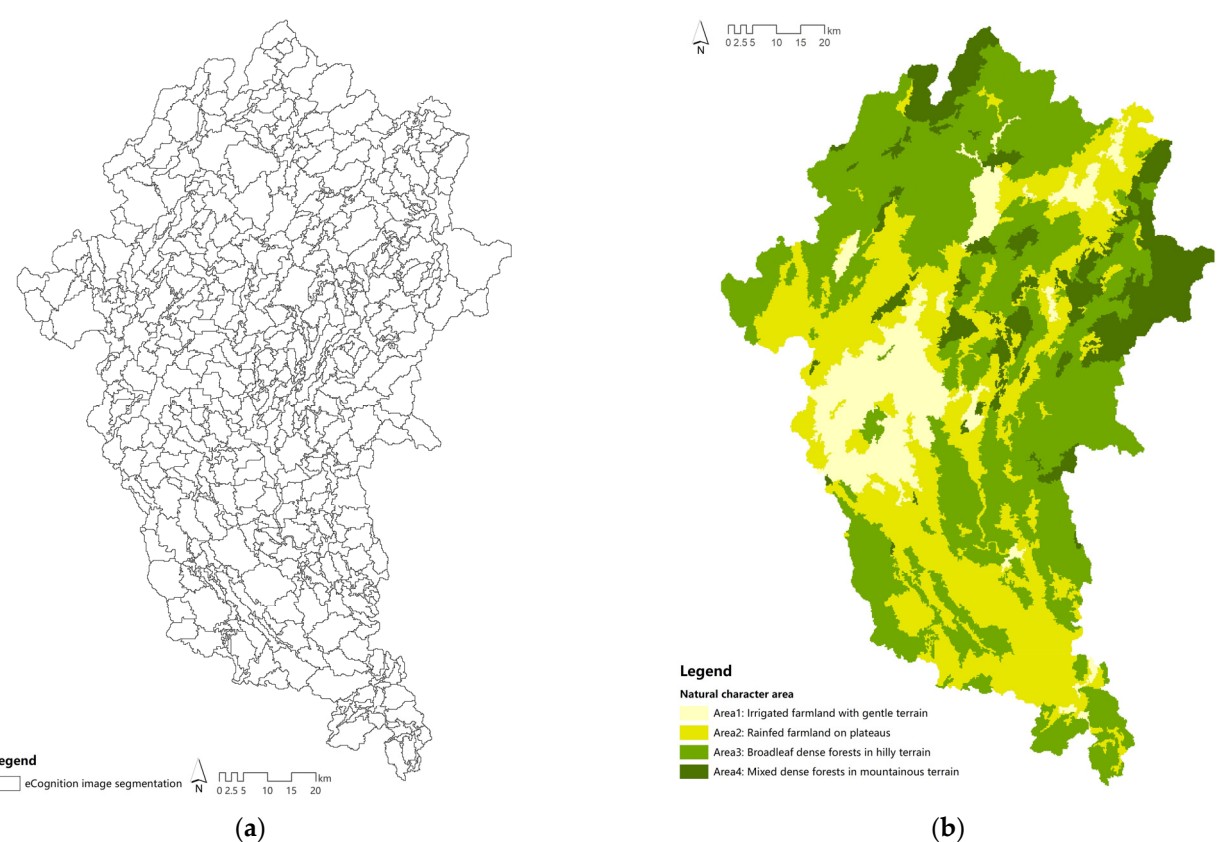

(**a**)                                                                 (**b**)

**Figure 7.** The process of natural character areas demarcation in the Li River Basin: (**a**) segmentation results using eCognition 9.0 software; (**b**) clustering results of natural character units.

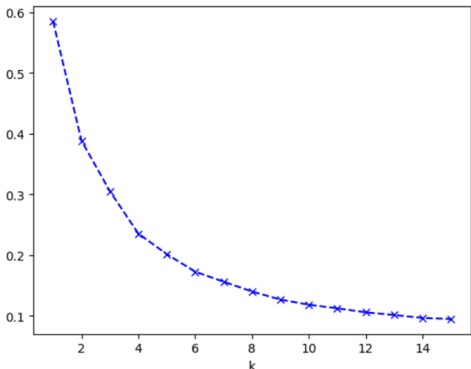

**Figure 8.** Elbow chart.

**Table 4.** Clustering silhouette coefficients for segmented regions.

| Cluster Number (K) | Silhouette Coefficient |
|:---:|:---:|
| 2 | 0.4855 |
| 3 | 0.5527 |
| 4 | 0.5839 |
| 5 | 0.5476 |
| 6 | 0.5124 |
| 7 | 0.5527 |
| 8 | 0.5141 |
| 9 | 0.5125 |
| 10 | 0.4956 |

Natural character regions exhibit the characteristic of "dominance of a single type, coexistence of multiple types". Analyzing the dominant landscape types in each region reveals the following (Table 5): (1) Landscape character Region 1 is dominated by type 2, accounting for 77.4%, followed by type 1 (15.41%), and types 3 and 4 with 5.31% and 1.97%, respectively. (2) Landscape character Region 2 has type 1, accounting for over 60%, followed by type 2 at 18.23%, and types 3 and 4 at 12.18% and 4.01%, respectively. (3) Landscape character Region 3 is predominantly type 3, accounting for 73.67%, with type 4 at 15.94%, and types 1 and 2 below 10%. (4) Landscape character Region 4 has types 4 and 3 accounting for 60.57% and 35.53%, respectively, with other types below 5%. According to the statistical results, the dominant landscape character types in Regions 1, 2, 3, and 4 are types 2, 1, 3, and 4, respectively. Analyzing the landscape elements of each type, the regions can be summarized as four areas: flat terrain with irrigated farmland, plateau terrain with rain-fed farmland, hilly terrain with broadleaf dense forest, and mountainous terrain with mixed dense forest.

**Table 5.** Percentage of area proportion and dominant types in natural character areas (%).

| Natural Character Area | Type 1 | Type 2 | Type 3 | Type 4 | Dominant Types |
|:---|:---:|:---:|:---:|:---:|:---:|
| Area 1: gently sloping irrigated farmland area | 15.41 | 77.40 | 5.31 | 1.97 | Type 2 |
| Area 2: plateau topography rain-fed farmland area | 65.57 | 18.23 | 12.18 | 4.01 | Type 1 |
| Area 3: hilly terrain broad-leaved dense forest area | 6.08 | 4.25 | 73.67 | 15.94 | Type 3 |
| Area 4: mountainous terrain mixed forest area | 4.29 | 1.36 | 33.53 | 60.57 | Type 4 |

### 3.3. Identifying Cultural Character Types and Areas

Cultural heritage units, traditional villages, and intangible cultural heritage are integral components of rural landscape heritage, collectively reflecting regional cultural characteristics. This study, based on the boundaries of cultural influence range, designates the cultural influence range of national heritage sources as the cultural core area, with the remaining area categorized as the cultural buffer zone. Based on the spatial distribution of

heritage sources and cultural content, six cultural categories and forty-three cultural units are established within the cultural core area, using national landscape heritage sources as the primary reference and considering other material heritage and intangible heritage. The core area comprises six cultural categories and twenty-two cultural units, while the buffer zone, with less distinct cultural characteristics, only includes cultural units, totaling twenty-one (Table 6; Figure 9).

**Table 6.** Cultural character areas in the Li River Basin.

| Cultural Area | Cultural Unit | Cultural Relic Site | Traditional Village | Intangible Heritage |
|---|---|---|---|---|
| 10 Military Water Conservancy Cultural Area | 11 Lingqu Unit | Lingqu*, Guyanguan, Yanguan Kiln Site, Tangjiadawu | Dong village *, Liutian village, Jianli village, Jiangxiping village, Yanguankou village, Liukouyan village | Guilin Dragon Boat Custom, Xing'an Rice Noodle Making Craft, Xing'an Dragon Boat Song, Mazi village |
| | 12 Qincheng Unit | Qincheng Site *, Shimaping Ancient Tomb Group | Rongliushang village *, Xiabei village, Yingshang village | |
| 20 Water Transportation Cultural Area | 21 Daxu Unit | Daxu Ancient Town, Fuziyan Site *, Liu Village Qin's Ancestral Hall and Stage, Mao village Virgin Temple, Luosheng Jiao Martyrs Cemetery, Xiong Village Ancient Architectural Complex | Lufang village *, Dabu village *, Mao village *, Taiping village *, Xiong village *, Longmen Village, Liu village | Guilin Fish Drum, Guilin Mountain and Water Legend, Fuli May 8 Folk Activity, Baisha "June 23" Festival, Li River Fishing Fire |
| | 22 Xingping Unit | Xingping Ancient Stage | Yu village | |
| | 23 Liugong Unit | - | Liugong village *, Fenglin village | |
| | 24 Longtan Unit | - | Langzi village *, Longtan village * | |
| | 25 Yulonghe Unit | Fuli Bridge, Xiangui Bridge, Yulong Bridge | Jiuxian village *, Yulongbao village * | |
| 30 Traditional Settlement Cultural Area | 31 Jiangtou Unit | Jiangtou Village Ancient Architectural Complex * | Jiangtou village * | Helang Song, Lingchuan Girl Festival, Guilin Round Bamboo Sliver Fan Making Craft, Lingchuan County Yang's Stone Carving Craft |
| | 32 Changgangling Unit | Changgangling Village Ancient Architectural Complex * | Changgangling village | |
| | 33 Ditang Unit | - | Ditang village *, Zhaiqing village *, Jinpen Village* | |
| | 34 Haiyang Unit | Tangjie Tomb, Sanyuan Pagoda, Datongmuwan Village Ancient Architectural Complex, Haiyang Temple | Dailou village *, Caiziyan village *, Shanwan village*, Datangbian village *, Muwan village *, Huangtutang village *, Jiangdong village, Fangtangling village, Qiaobian village | |
| | 35 Shuiyuantou Unit | Shuiyuantou Village Ancient Architectural Complex * | Shuiyuantou village *, Dalukou village, Tangkoutian village, Yanmenqian village, Shizhu village | |
| | 36 Bangshang Unit | Bangshang Village Ancient Architectural Complex * | Bangshang village, Zhongshanping village * | |

**Table 6.** *Cont.*

| Cultural Area | Cultural Unit | Cultural Relic Site | Traditional Village | Intangible Heritage |
|---|---|---|---|---|
| 40 Ethnic Minority Cultural Area | 41 Xizhou Unit | - | Xizhou Zhuangzhai village *, Sanxiandong village | Xing'an Yao Embroidery, Lingchuan Lantan Yao Papermaking Craft, Dajingzhai Yao Tea Making Craft |
| | 42 Qingshanwan Unit | - | Qingshanwan village | |
| | 43 Laozhai Unit | - | Xinzhai village *, Laozhai village * | |
| 50 Red Revolutionary Culture Area | 51 Jieshou Unit | Guanghuapu Ambush Battle Site *, Jieshou Ferry Site *, Jieshou Sanguan Hall *, Jieshou Jielong Bridge, Guanghuapu Red Army Martyrs Cemetery, Jieshou Ancient Tomb Group, Shuangzaotian Ancient Tomb Group, Jiandidiangong Bridge, Wen's Ancestral Hall | Chaohuangdian village, Lijia village, Xiazaiyan village, Jiangnan village, Qukoulao village | |
| | 52 Qianjiasi Unit | Qianjiasi Red Army Slogan Building | - | |
| | 53 Luxi Unit | Former Site of Military Supplies Transfer Station | Luxi village * | |
| | 54 Yangtang Unit | Former Site of Yangtang Flying Tiger Command, Dayan Site * | - | |
| 60 Suburban Folk Culture Area | 61 Yanshan Unit | Ma Junwu Tomb, Dagangbu Tang's Manor, Zhuyuan village Han Tomb Group, Yanshan Park, Science Museum, Republic of China Guangxi University Branch Site, Miaoyantang Cave Site | Dagangbu village *, Jiu village *, Shanwei village, Xinglong village, Liangfengxia village | Bandeng Dragon, Guilin Brand, Lingchuan Grass Dragon Dance, Lingchuan Colorful villages |
| | 62 Hengshan Unit | Hengshan Chen's Ancestral Hall and Stone Carvings | Hengshan village * | |
| 70 Watershed Cultural Buffer Zone | 71–721 Unit | - | - | - |

* National rural landscape heritage source.

*3.4. Identification and Analysis of Rural Landscape Heritage Character Areas*

Overlaying natural and cultural character regions (Figure 10a) yields heritage character regions, with a nomenclature following the "natural character type number-cultural character unit number" rule. A total of 268 landscape character areas were identified (Figure 10b). The results reveal the overall landscape differences in the Li River Basin and its rural landscape heritage: In terms of natural characteristics, the northwestern and southeastern areas have higher elevations, dominated by broadleaf and coniferous forests in hilly and mountainous terrains. In contrast, the central and southern regions are lower, characterized by plains and tableland landscapes with predominant agricultural fields. Regarding cultural characteristics, the northern upstream area, influenced by the ling Canal, exhibits rich military camp cultural elements. In the northwestern and south-

eastern regions, villages are constructed in hilly and mountainous areas, showcasing a strong tradition of village cultural heritage preservation. In the southern area, heritage near the main and major tributaries, such as the Yulong River, is dominated by a waterway transportation culture.

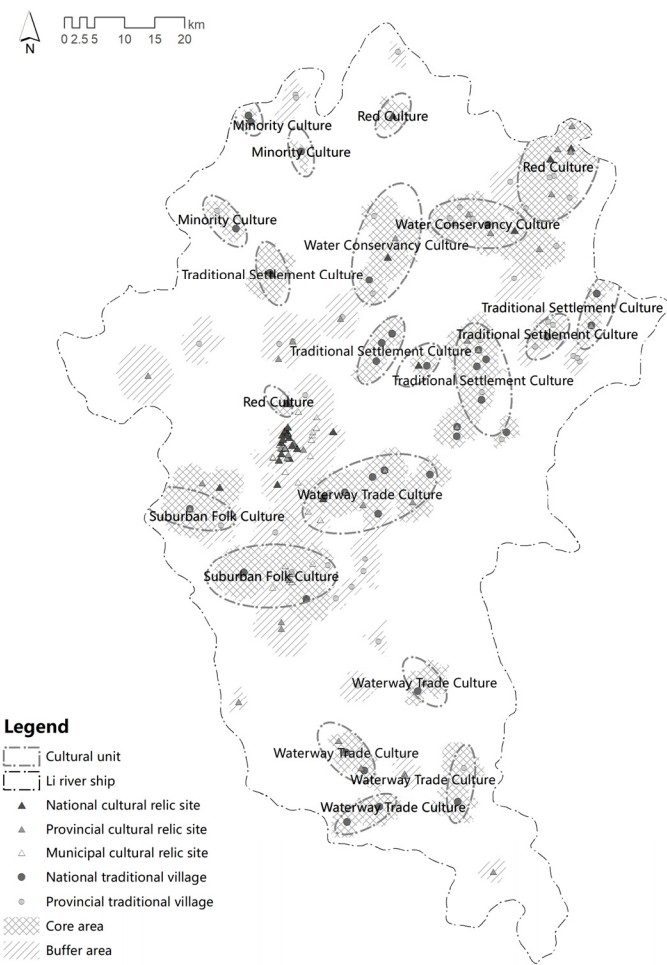

**Figure 9.** Cultural character areas in the Li River Basin.

This study utilizes the cultural character regions of heritage sources as the scope for rural landscape heritage. Through overlay, analysis, and adjustment, a total of 150 landscape character areas were identified (Figure 11), including four natural character areas, seven cultural character areas, and forty-three heritage character units. Among the four natural character types, tableland landscapes with rain-fed agricultural fields are the largest, covering an area of 1180.24 km$^2$ (49.86%), followed by plains with irrigated agricultural fields, covering 663.29 km$^2$ (28.02%). Hilly terrains with broadleaf forests and mountainous terrains with mixed coniferous forests cover 400.84 km$^2$ (16.93%) and 122.73 km$^2$ (5.18%), respectively. Concerning cultural character areas, Zones 1–6 represent the core areas, while Zone 7 serves as the buffer area. The buffer area is the largest, covering 817.43 km$^2$ (34.53%). Military and hydraulic cultural areas, water transportation cultural areas, traditional settlement cultural areas, Red Army resistance cultural areas, and suburban folk cultural areas are of similar sizes comprise 235.55 km$^2$ (9.95%), 393.07 km$^2$ (16.61%), 329.24 km$^2$ (13.91%), 291.22 km$^2$ (12.30%), and 259.94 km$^2$ (10.98%), respectively. The smallest is the ethnic minority cultural area, with an area of 40.65 km$^2$ (34.53%) (Figure 12). Overall, rural landscape heritage in the Li River Basin is concentrated in the upper and middle reaches of the basin, representing a region where diverse natural landscape characteristics coexist with a blend of multiple cultures.

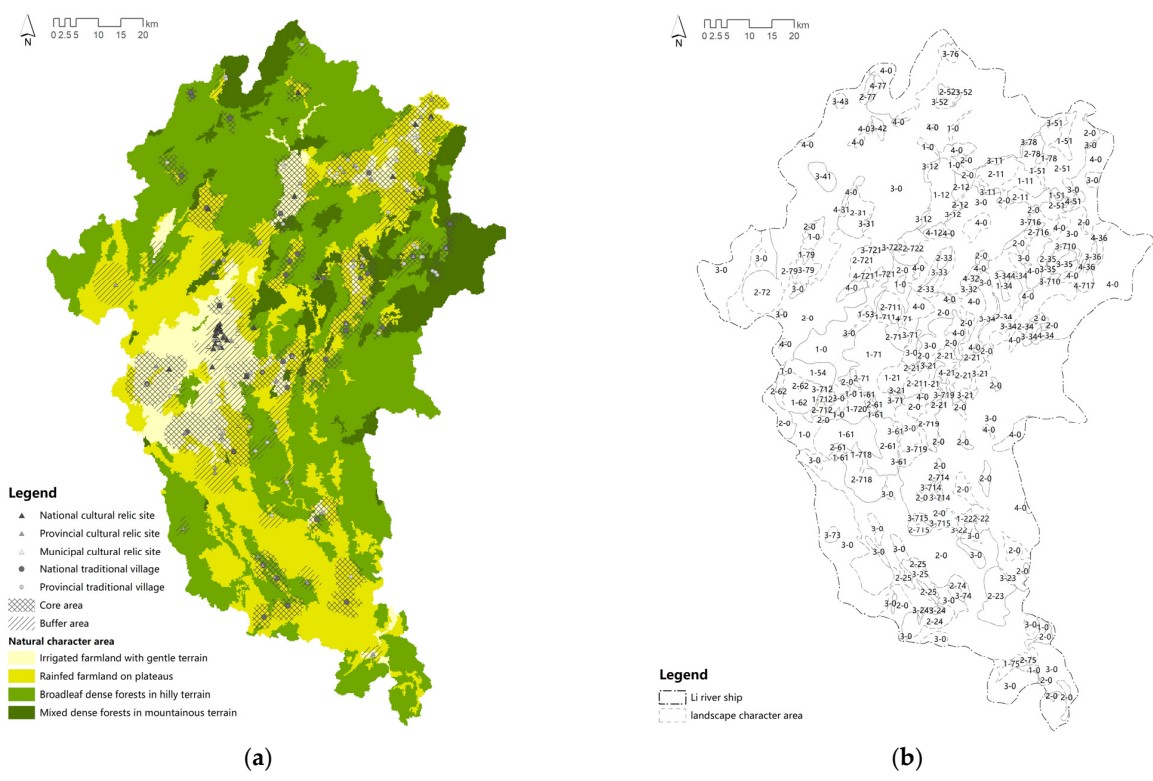

**Figure 10.** Rural landscape character areas in the Li River Basin: (**a**) overlay analysis; (**b**) rural landscape character areas.

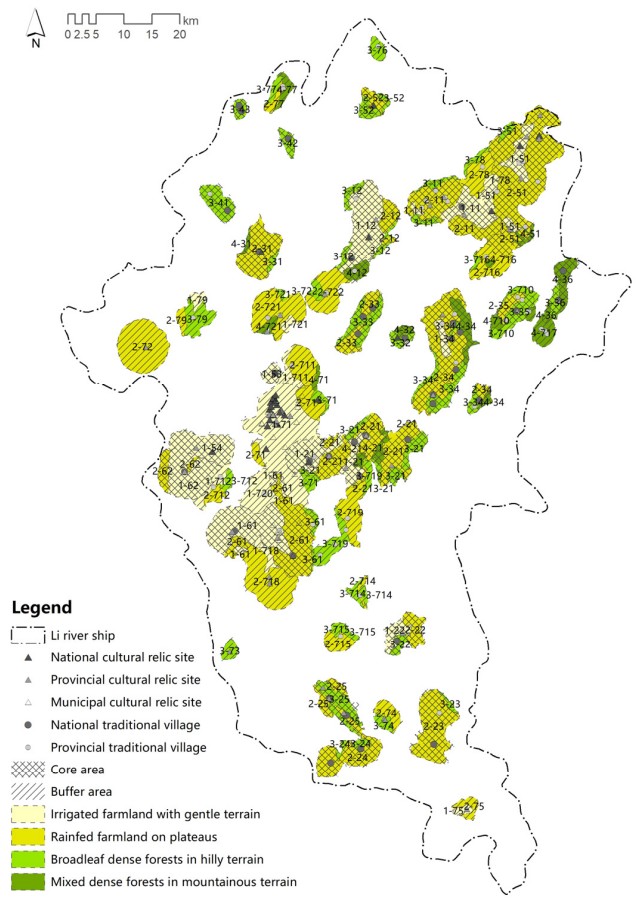

**Figure 11.** Rural landscape heritage character areas in the Li River Basin.

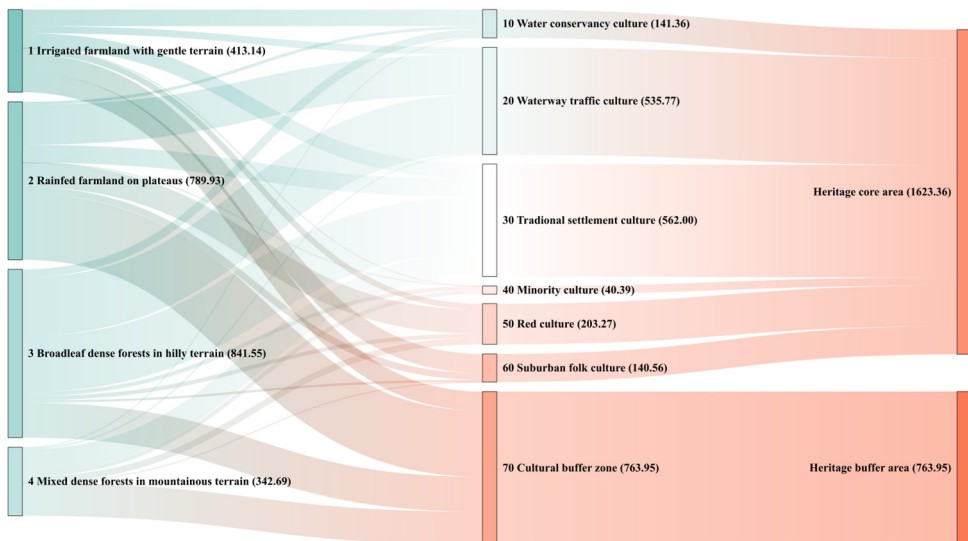

**Figure 12.** Information on the rural landscape heritage character area of the Li River Basin.

## 4. Discussion

### 4.1. Identification of Rural Landscape Heritage Character

In this study, we introduced the Landscape Character Assessment (LCA) method to identify the characteristics of rural landscape heritage in the Li River Basin. The process is primarily divided into the identification of natural characteristics and cultural characteristics.

At the natural level, to enhance the applicability and efficiency of identifying heritage natural characteristics, a new process of character unit clustering was introduced. This addition aims to improve the application scenarios and efficiency of recognizing natural characteristics of heritage. In the existing research on landscape character identification, clustering and automatic segmentation methods are widely used [42]. These methods often involve the clustering of raster data in target areas based on elevation, topography, land cover, soil, etc. Subsequently, image segmentation software is used to partition the clustered raster data [49]. However, in practical applications, it was observed that the mapped images after raster clustering exhibit a high degree of fragmentation. This hampers the swift identification of the natural character types associated with heritage. It is evident that a gap exists between the directly clustered type zone map and the analysis of the natural characteristics of heritage sources. To address this, and to enhance both efficiency and applicability, the process of natural character identification was divided into four steps: raster clustering, image segmentation, unit clustering, and integrated identification. The unit clustering step is an addition to the existing methods. After raster clustering, natural units were established through image segmentation, and a second clustering based on these units was performed to create landscape character zones. The addition of the unit clustering step aims to address the issue of image fragmentation after raster clustering. This not only enhances the efficiency of the identification process but also broadens the application scenarios of character recognition.

At the cultural level, we established a cultural data structure with a focus on rural landscape characteristics to identify the characteristics of rural landscape heritage in the Li River Basin. Currently, the LCA guidelines in the United Kingdom provide a corresponding list of cultural factors, including land use, settlement patterns, building types and materials, farming patterns and types, urban morphology, land ownership, and historical dimensions. However, these indicators are not only challenging to apply at a macro scale but are also not comprehensive enough for rural landscapes. Accordingly, based on the definition of rural landscapes as heritage in the "Criteria for Rural Landscape as Heritage" document, we established a cultural factor data structure. This structure includes both material heritage and non-material heritage in rural areas [4]. Material heritage includes cultural heritage

sites and traditional villages, while non-material heritage comprises intangible cultural heritage. These heritage sources have been evaluated by various levels of the Chinese government and possess high cultural value and distinctive local characteristics. They can provide a more comprehensive reflection of the cultural characteristics of rural landscapes.

*4.2. Protection of Rural Landscape Heritage Areas*

The Lijiang River Basin is a region with distinct geographical characteristics and a typical area of multi-cultural coexistence. Through the identification of landscape character areas, it is possible to integrate natural and cultural elements to determine the spatial characteristics of heritage areas. This facilitates the understanding of the landscape spatial traits and connections among heritage sources within the region, providing a foundation for regional protection and management of heritage. Rural landscapes, to a certain extent, represent the coordinated construction of nature and culture [50]. Simultaneously, the establishment of protective zones and the formulation of differentiated measures are essential approaches for managing heritage [51]. This is conducive to enhancing the scientific and effective protection of heritage. Based on the identification of the landscape characteristics of the Li River Basin above, combined with the field investigation and assessment (Appendix A), heritage protection strategies can be proposed from the natural and cultural levels.

In terms of protecting natural characteristics, based on the classification of natural character types, two main categories are identified: plain and plateau farmland type, and hill and mountain dense forest type. Corresponding protection methods and strategies are proposed to assist in achieving comprehensive protection and differentiated management of rural landscape heritage in the Li River Basin. For the plain and plateau farmland areas, efforts should be made to strengthen the connections between rural heritage areas, establishing a heritage network centered around agriculture and rural resources. Regarding the hill and mountain dense forest areas, a balance between development and ecological preservation should be maintained. This involves minimizing the ecological impact of land development and ensuring the protection of landscape resources, flora and fauna, and biodiversity.

Regarding the protection of cultural characteristics, a hierarchical protection approach is proposed based on the distinction between core and buffer zones. The core area has a rich variety of heritage types, including traditional architectural complexes, traditional cultivation spaces, local folk culture, etc. All types of heritage resources should be fully integrated within the region, and scientific planning and rational organization should be carried out to promote a more reasonable layout and balanced development of the regional society. The buffer zone, with relatively weaker cultural influence, serves as both an extension of the heritage area and a transition between heritage areas. Therefore, in terms of protection, efforts should be intensified to enhance cultural exchange and social connections with the core heritage areas. This contributes to establishing a comprehensive heritage network by providing a connecting function between different heritage areas.

**5. Conclusions**

Rural landscape heritage should not only focus on the heritage itself and its buffer zones but also emphasize understanding landscape characteristics from the overall perspective of regions and units. The identification and zoning of landscape characteristics are beneficial for preserving and managing the landscape diversity and uniqueness of heritage areas. This paper is divided into four steps: selection and spatial scope identification of rural landscape heritage, identification of natural character areas, identification of cultural character areas, and identification and analysis of character areas of rural landscape heritage.

The conclusions of this study are as follows: (1) The selection cultural relic units, traditional villages, and intangible cultural heritage as sources of rural landscape heritage was carried out by utilizing the Minimum Cumulative Resistance model (MCR) to calculate

the spatial scope of rural landscape heritage. (2) Clustering and automatic partition methods were employed to classify the Li River Basin into four types of natural character areas. (3) Cultural core areas and buffer areas were determined based on the heritage source hierarchy and cultural features. (4) By overlaying heritage spatial ranges, natural character areas, and cultural character areas, 2 levels of heritage areas, 7 types of heritage cultural areas, and 43 heritage character units were obtained. This study, through the identification of heritage characteristics, has delineated heritage areas. In the future, it can be extended to various scales based on the identification results by constructing cultural heritage corridors and characteristics in a macroscopic perspective, analyzing in depth the spatial structure characteristics and relationships between heritages from a mesoscopic perspective, and reconsidering, from a microscopic perspective, the delineation of heritage scope based on landscape characteristics.

Rural landscape heritage is an organic system where "heaven, earth, and humanity" coexist symbiotically. Establishing characteristic areas for rural landscape heritage in the Li River Basin can define the spatial scope of heritage landscapes, supporting the overall protection of heritage landscapes. This has significant value and meaning for the sustainable conservation and inheritance of natural and cultural resources in rural areas.

**Author Contributions:** Conceptualization, W.Z. and Z.H.; methodology, Z.H.; software, Z.H.; validation, Z.H.; formal analysis, Z.H.; investigation, W.C.; resources, Y.C.; data curation, Z.H.; writing—original draft preparation, Z.H.; writing—review and editing, W.Z. and S.Z.; visualization, Z.H.; supervision, W.Z.; project administration, W.Z.; funding acquisition, W.Z. All authors have read and agreed to the published version of the manuscript.

**Funding:** This research was funded by the Guangxi Key Research and Development Plan project, grant number AB22080057; the Guangxi Key Research and Development Plan project, grant number AB23026053.

**Institutional Review Board Statement:** Not applicable.

**Informed Consent Statement:** Not applicable.

**Data Availability Statement:** Data are contained within the article.

**Conflicts of Interest:** The authors declare no conflicts of interest.

## Appendix A

| Aerial view | LOCATION | CHARACTER AREA | | PROTECTION STRATEGY | |
|---|---|---|---|---|---|
| | Yanguan Village | (10) Military Water Conservancy Cultural Area | | Connect nodes and strengthen interactions | |
| | | **CHARACTER AREA NUMBER** | | | |
| | | 1-11 | | | |
| | **PERCEPTION** | | | | |
| | SECURITY | COMFORTABLE | SAFE | UNSETTLING | THREATENING |
| | STIMULUS | BLAND | INTERESING | INSPIRING | INVIGORATING |
| | SERENITY | INACCESSIBLE | REMOTE | VACANT | PEACEFUL |
| | PLEASURE | UNPLEASANT | PLEASANT | ATTRACTIVE | BEAUTIFUL |
| Ling Canal | **VISUAL ASSESSMENT** | | | | |
| | PATTERN | DOMINAT | STRONG | BROKEN | WEAK |
| | FORM | AGGLOMERATE | RIBBON | DACTYLIFORM | |
| | SCALE | INTIAMTE | SMALL | MEDIUM | LARGE |
| | TEXTURE | SMOOTH | TEXTURED | ROUGH | VERY ROUGH |
| | COLOUR | SINGLE | MUTED | COLOURFUL | GARISH |
| | MOVEMENT | STILL | CALM | ACTIVE | FRENETIC |
| | BALANCE | BALANCED | UNBALANCED | | |
| | STRUCTURE | RANDOM | REGULAR | FORMAL | |

**Figure A1.** Field survey sheet employed at military water conservancy cultural area.

| Aerial view | LOCATION | CHARACTER AREA | | PROTECTION STRATEGY | |
|---|---|---|---|---|---|
| | Liugong Village | (20) Water Transportation Cultural Area | | Connect nodes and strengthen interactions | |
| | | **CHARACTER AREA NUMBER** | | | |
| | | 2-23, 3-23 | | | |
| | | **PERCEPTION** | | | |
| | SECURITY | COMFORTABLE | SAFE | UNSETTLING | THREATENING |
| | STIMULUS | BLAND | INTERESING | INSPIRING | INVIGORATING |
| | SERENITY | INACCESSIBLE | REMOTE | VACANT | PEACEFUL |
| | PLEASURE | UNPLEASANT | PLEASANT | ATTRACTIVE | BEAUTIFUL |
| Local graph | | **VISUAL ASSESSMENT** | | | |
| | PATTERN | DOMINAT | STRONG | BROKEN | WEAK |
| | FORM | AGGLOMERATE | RIBBON | DACTYLIFORM | |
| | SCALE | INTIAMTE | SMALL | MEDIUM | LARGE |
| | TEXTURE | SMOOTH | TEXTURED | ROUGH | VERY ROUGH |
| | COLOUR | SINGLE | MUTED | COLOURFUL | GARISH |
| | MOVEMENT | STILL | CALM | ACTIVE | FRENETIC |
| | BALANCE | BALANCED | UNBALANCED | | |
| | STRUCTURE | RANDOM | REGULAR | FORMAL | |

**Figure A2.** Field survey sheet employed at water transportation cultural area.

| Top view | LOCATION | CHARACTER AREA | | PROTECTION STRATEGY | |
|---|---|---|---|---|---|
| | Changgangling Village | (30) Traditional Settlement Cultural Area | | Protect the heritage and control development | |
| | | **CHARACTER AREA NUMBER** | | | |
| | | 3-32, 4-32 | | | |
| | | **PERCEPTION** | | | |
| | SECURITY | COMFORTABLE | SAFE | UNSETTLING | THREATENING |
| | STIMULUS | BLAND | INTERESING | INSPIRING | INVIGORATING |
| | SERENITY | INACCESSIBLE | REMOTE | VACANT | PEACEFUL |
| | PLEASURE | UNPLEASANT | PLEASANT | ATTRACTIVE | BEAUTIFUL |
| Aerial view | | **VISUAL ASSESSMENT** | | | |
| | PATTERN | DOMINAT | STRONG | BROKEN | WEAK |
| | FORM | AGGLOMERATE | RIBBON | DACTYLIFORM | |
| | SCALE | INTIAMTE | SMALL | MEDIUM | LARGE |
| | TEXTURE | SMOOTH | TEXTURED | ROUGH | VERY ROUGH |
| | COLOUR | SINGLE | MUTED | COLOURFUL | GARISH |
| | MOVEMENT | STILL | CALM | ACTIVE | FRENETIC |
| | BALANCE | BALANCED | UNBALANCED | | |
| | STRUCTURE | RANDOM | REGULAR | FORMAL | |

**Figure A3.** Field survey sheet employed at traditional settlement cultural area.

| Top view | LOCATION | CHARACTER AREA | | PROTECTION STRATEGY | |
|---|---|---|---|---|---|
| | Laozhai Village | (40) Ethnic Minority Cultural Area | | Protect the heritage and control development | |
| | | **CHARACTER AREA NUMBER** | | | |
| | | 3-43 | | | |
| | | **PERCEPTION** | | | |
| | SECURITY | COMFORTABLE | SAFE | UNSETTLING | THREATENING |
| | STIMULUS | BLAND | INTERESING | INSPIRING | INVIGORATING |
| | SERENITY | INACCESSIBLE | REMOTE | VACANT | PEACEFUL |
| | PLEASURE | UNPLEASANT | PLEASANT | ATTRACTIVE | BEAUTIFUL |
| Aerial view | | **VISUAL ASSESSMENT** | | | |
| | PATTERN | DOMINAT | STRONG | BROKEN | WEAK |
| | FORM | AGGLOMERATE | RIBBON | DACTYLIFORM | |
| | SCALE | INTIAMTE | SMALL | MEDIUM | LARGE |
| | TEXTURE | SMOOTH | TEXTURED | ROUGH | VERY ROUGH |
| | COLOUR | SINGLE | MUTED | COLOURFUL | GARISH |
| | MOVEMENT | STILL | CALM | ACTIVE | FRENETIC |
| | BALANCE | BALANCED | UNBALANCED | | |
| | STRUCTURE | RANDOM | REGULAR | FORMAL | |

**Figure A4.** Field survey sheet employed at ethnic minority cultural area.

| Aerial view | LOCATION | CHARACTER AREA | | PROTECTION STRATEGY | |
|---|---|---|---|---|---|
| | Luxi Village | (50) Red Revolutionary Culture Area | | Maintain the culture and activate spaces | |
| | | CHARACTER AREA NUMBER | | | |
| | | 1-53 | | | |
| | | PERCEPTION | | | |
| | SECURITY | COMFORTABLE | SAFE | UNSETTLING | THREATENING |
| | STIMULUS | BLAND | INTERESING | INSPIRING | INVIGORATING |
| | SERENITY | INACCESSIBLE | REMOTE | VACANT | PEACEFUL |
| | PLEASURE | UNPLEASANT | PLEASANT | ATTRACTIVE | BEAUTIFUL |
| Local graph | | VISUAL ASSESSMENT | | | |
| | PATTERN | DOMINAT | STRONG | BROKEN | WEAK |
| | FORM | AGGLOMERATE | RIBBON | DACTYLIFORM | |
| | SCALE | INTIAMTE | SMALL | MEDIUM | LARGE |
| | TEXTURE | SMOOTH | TEXTURED | ROUGH | VERY ROUGH |
| | COLOUR | SINGLE | MUTED | COLOURFUL | GARISH |
| | MOVEMENT | STILL | CALM | ACTIVE | FRENETIC |
| | BALANCE | BALANCED | UNBALANCED | | |
| | STRUCTURE | RANDOM | REGULAR | FORMAL | |

**Figure A5.** Field survey sheet employed at red revolutionary cultural area.

| Aerial view | LOCATION | CHARACTER AREA | | PROTECTION STRATEGY | |
|---|---|---|---|---|---|
| | Hengshan Village | (60) Suburban Folk Culture Area | | Maintain the pattern and activate spaces | |
| | | CHARACTER AREA NUMBER | | | |
| | | 1-62, 2-62 | | | |
| | | PERCEPTION | | | |
| | SECURITY | COMFORTABLE | SAFE | UNSETTLING | THREATENING |
| | STIMULUS | BLAND | INTERESING | INSPIRING | INVIGORATING |
| | SERENITY | INACCESSIBLE | REMOTE | VACANT | PEACEFUL |
| | PLEASURE | UNPLEASANT | PLEASANT | ATTRACTIVE | BEAUTIFUL |
| Local graph | | VISUAL ASSESSMENT | | | |
| | PATTERN | DOMINAT | STRONG | BROKEN | WEAK |
| | FORM | AGGLOMERATE | RIBBON | DACTYLIFORM | |
| | SCALE | INTIAMTE | SMALL | MEDIUM | LARGE |
| | TEXTURE | SMOOTH | TEXTURED | ROUGH | VERY ROUGH |
| | COLOUR | SINGLE | MUTED | COLOURFUL | GARISH |
| | MOVEMENT | STILL | CALM | ACTIVE | FRENETIC |
| | BALANCE | BALANCED | UNBALANCED | | |
| | STRUCTURE | RANDOM | REGULAR | FORMAL | |

**Figure A6.** Field survey sheet employed at suburban folk cultural area.

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
