# Peer review of "Identifying Rural Landscape Heritage Character Types and Areas: A Case Study of the Li River Basin in Guilin, China"

_sustainability, doi:10.3390/su16041626_

Round 1

Reviewer 1 Report

Comments and Suggestions for Authors

In this paper, the authors investigated a relatively large area that had previously been the subject of many studies, Especially due to its natural and cultural features. Therefore, it was not easy for the authors to find a new research cpntribution. Originalty in the work exists in the sense of overlapping locations with significant natural and cultural heritage in terms od landscape. A lot of work has been done to analyse natural features through GIS methods as well as cultural heritage through other methods. 1. In addition to that, the work also has numerous shortcomings related to the very context that authors chose: nature, culture, heritage (disputed places are listed in the reviewed text). 2. In the introductory part, among other things, the authors did not excessively refer to the previous researches of the same region that they dealing with, even though there are very similar researches (see in the text). 3. There is no Literature Review chapter even though the region has been extensively researched previously. 4. In several places in the text, culture and heritage are considered separately, and in one place they are shown as one (in the table). I believe that culture is part of the heritage, so it should not be separated. 5. Some text segments from the Discussion belong to the Methodology and some belong to the Conclusion. 6. In addition, some minor corrctions were suggested to the author.

Author Response

Dear Reviewer,

I hope this letter finds you well. I would like to express my gratitude to you for the insightful comments and suggestions provided during the review process for our manuscript titled "Integration of Natural and Cultural Characters in Assessing Rural Landscape Heritage: A Case Study of the Li River Basin." We appreciate the time and effort put forth in evaluating our work.

We made several crucial improvements to our manuscript. First, the overview section of the Li River Basin has been rewritten, and the introduction now more clearly describes the importance of the Li River basin and the sustainable challenges it faces, allowing readers to gain a clearer understanding of the purpose of the study. Secondly, the methods section underwent substantial revisions to better convey the comprehensiveness of natural and cultural elements in the identification of rural landscape heritage characteristics. Thirdly, it is clearly proposed to include the Lijiang River basin as a "South China Karst" World Natural Heritage site and a national scenic spot in China. Fourthly, other revisions were made based on your peer-review comments, as detailed in the attached document.

We believe these revisions have significantly strengthened the manuscript. We hope the changes address the concerns raised during the review process. Our point-by-point responses to the comments and suggestions are set out in the Appendix.

Thank you for considering our revised manuscript. We look forward to your feedback and guidance.

Sincerely,

Zizhen Hong

Reviewer 2 Report

Comments and Suggestions for Authors

The article presents itself as a valuable and innovative study that addresses the significant issue of rural landscape heritage. The authors' proposal to combine different Landscape Character Assessment (LCA) methods for identifying heritage characteristics and areas seems to be a suitable solution for the current challenges related to landscape character homogenization and unclear protection boundaries.

The chosen research area, the Li River Basin in Guilin, China, appears to be well-suited, allowing for the translation of the proposed methodology into a specific context. The three-step method for identifying heritage characteristics and areas, involving natural character zoning, cultural character zoning, and heritage character zoning, appears to be structurally coherent and logical.

The paper also presents the application of advanced methods such as clustering, automatic division, and the Minimum Cumulative Resistance (MCR) model, indicating the thoroughness and sophistication of the analysis. It is worth emphasizing that the proposed tools could potentially be useful in practice, and their application may contribute to a better understanding and protection of landscape heritage.

One aspect that could enhance the article is the inclusion of hypotheses with which the researchers approached the study. This addition, particularly at the end of the introductory paragraph, would provide readers with a clearer understanding of the study's goals and expectations.

The use of integration and overlay analysis of character layers further enriches the presented methodology, leading to the identification of specific heritage grade areas and cultural heritage areas. The effectiveness in identifying areas with varying degrees of heritage and cultural heritage areas is noteworthy.

The conclusions drawn from the article appear practical and may serve as a valuable source of information for decision-makers involved in the preservation of landscape heritage at both local and global levels. Therefore, I recommend accepting the article in its proposed form, with the suggested addition of hypotheses for a more comprehensive introduction. This adjustment could strengthen the overall clarity and impact of the research.

Author Response

(The authors gave the same response as above.)

Round 2

Reviewer 1 Report

Comments and Suggestions for Authors

The authors approached the response to the review very systematically. They answered all suggestions, questions and doubts.

- The key suggestion was that heritage is part of culture and vice versa. The authors managed to change the text and explain that they see landscape and culture as the heritage of a region. It is much clearer to me now. Based on that, the scheme on page 5 was also changed.

- The map on page 3 is a great contribution to this work and brings the explored area closer to every reader in a simple way.

- The authors in the new version have paragraphs that correspond to the subheadings to which they belong (they moved the text on methodology within the Methods chapter)

The corrected version of the work is an example of a complex regional-geographical analysis and synthesis that can serve as a basis for works of a similar spatial scope (macroregion). I am pleased to propose the publication of this paper.